# Broadcast Event-Triggered Control Scheme for Multi-Agent Rendezvous Problem in a Mixed Communication Environment

**Nohaidda Sariff [1] and Zool Hilmi Ismail [2,*]**

[1] Department of Electrical and Electronics Engineering, Faculty of Engineering, Al-Madinah International University, Kuala Lumpur 57100, Malaysia; nohaidda.sariff@mediu.edu.my
[2] Centre for Artificial Intelligence and Robotics, Universiti Teknologi Malaysia, Jalan Sultan Yahya Petra, Kuala Lumpur 54100, Malaysia
* Correspondence: zool@utm.my; Tel.: +60-199-816-001

**Featured Application: In this work, a broadcast event-triggered control approach is used to effectively solve the rendezvous problem and to improve the performance of a multi-agent system in a mixed communication environment.**

**Abstract:** This paper addresses the communication issue encountered by a hybrid controller when finding consensus in terms of the rendezvous target point in a broadcast and communication environment. This issue may result in a high level of computation and the utilization of agent resources when a continuous communication is required by agents to meet convergence requirements. Thus, an event-triggered system was integrated into the design of a broadcast and distributed consensus linear controller using the simultaneous perturbation stochastic algorithm (SPSA). The agent's movement towards the rendezvous point is based on the broadcast value, whereas the next agent's state position depends on the distributed local controller output. The communication error obtained during communication between the agent and neighbors is only added to the gradient approximation error of the SPSA if the event-triggered function is violated. As a result, in our model, the number of channel utilizations was lower and the agents' performances were preserved. The efficiencies and effectiveness of the proposed controller have been compared with the traditional sampling broadcast time-triggered (BTT) approach. The time and iterations required by the broadcast event-triggered (BET) system were less than 40.42% and 21% on average as compared to BTT. The trajectory was not the same—the BET showed scattered movements at the initial stage, whereas BTT showed a linear movement. In terms of the number of channels, 28.91% of channels were preserved during the few hundred iterations. Consequently, a variety of hybrid controllers with event-triggered mechanisms can be proposed for other multi-agent motion coordination tasks.

**Keywords:** multi-agent system; consensus control; broadcast; stochastic approximation; event-triggered

## 1. Introduction

Multi-agent robot research has been expanding due to the effectiveness, robustness, flexibility and operational efficiency involved in accomplishing tasks with many agents, compared to the use of a single agent [1]. The advantages of load-sharing, enable complex tasks to be simplified for agents, which has allowed cooperative multi-agent robot research to be more actively explored [2–4]. The multi-agent system is specifically designed to work in hazardous environments and in very limited spaces, which are impossible for humans to reach. Multi-agent systems have applications in medicine and nanotechnology [5]—magnetic [6] robots have been used to send medicine directly to human organs, planetary rovers have been used to accomplish missions on other planets [7,8], inspection robots have been used to inspect and clean the pipelines in oil and gas industries [9,10] and researchers have explored the use of unmanned aerial vehicles and swarm robot applications [11–13]. Due to the importance of these

multi-agent systems in aiding human activities, various issues related to the cooperation and coordination of multi-agent systems have been researched and discussed up to the present day.

The analysis and direction of cooperative multi-agent systems have been reviewed and discussed by several researchers [2,14] to demonstrate the significance and impact of multi-agent coordination research. Several issues of multi-agent coordination and control, such as formation [15], consensus [16], containment [17], task allocation [18], path planning [19] and rendezvous [20], have been highlighted by previous researchers. Among them, consensus, which consists of an agreement between agents to reach a certain quality of interest, has attracted a large amount of research. Most of this consensus research has focused on positional consensus or velocity, applied either to homogeneous or heterogeneous agents. Consensus issues can also be categorized into leaderless consensus [16], leader–follower consensus [21], output consensus problems [22] or positional consensus [23]. In parallel with recent wireless network technologies, consensus research is in demand when there are various topology networks [1] that exist to present the connection between the agent and neighbors. The flow of information, which can either be directed or undirected, one-way or two-way, and distributed/one-to-one or centralized/broadcast, that represents the communication between multi-agent makes the consensus issue more challenging.

Control and communication issues involved in obtaining an effective consensus system have also attracted the attention of many researchers. Finding a consensus for formation, tracking and rendezvous control applications is challenging for multi-agent systems. For formation, model predictive control has been proposed for a unmanned aerial vehicles (UAV), leader-following system [13] and vector field path planning has been designed for multiple decentralized UAVs without a leader [11]. In addition, the use of a novel state observer and sliding mode control for heterogeneous systems [24] and adaptive control laws for tracking between leader–follower systems [25] are examples of consensus controllers for tracking applications. Meanwhile, for consensus to a rendezvous point, which also known as positional consensus, the focus has been on the design of an effective consensus controller, which can be used in various applications. Models using linear programming with random perturbation [16], a constructed intermediate attitude system for unicycle-type vehicles [26] and distributed static and dynamic control [27], discrete-time double integrator control [28] and broadcast controllers [17] are among the proposed controllers. Additionally, some other researchers focus on the communication issue problem in relation to consensus and rendezvous. For instance, Dong Yi [29] developed a leader–follower robot by proposing an observer that was used to check the independent triggering of event-triggered (ET), whereas Bing Xian Mu et al. [20] focused on aperiodic detection with a communication delay for the application of multiple two-wheeled mobile robots.

Other than control, the realization of consensus depends on the effectiveness of the communication system [14,30]. Issues like time delays [31,32], disturbances [33,34] and limited resources [35,36] have been discussed. Xing et al. [32] solved the time delay problem using the parametric algebraic Riccati equation, whereas Noorbakhsh et al. [37] proposed a heuristic dynamic programming method. For disturbances in communication transmission, Cheng et al. [38] solved the issue of hybrid multi-agent systems with unknown disturbances, whereas Zhang et al. [39] proposed a solution for unknown external disturbances in the tracking system. Due to the limitations of communication resources embedded in the multi-agent controller, event-based control has been applied to solve the communication issue during the consensus process [35,36,40,41].

The idea of having an event-triggered control was proposed in order to reduce the frequency of sampling when the agent receives the input signal from the sensor to be sent to the controller. By reducing the transmission of the signal, the computation and communication resources can be reduced to guarantee the practicality of the controller [28,33,34,42–44]. Due to the significance of event-triggering in multi-agent consensus systems, event-based control has been integrated into consensus control systems by means of sliding mode control (SMC) [45], a fuzzy logic and back stepping technique [46], model predictive control (MPC) [47], distributed control [20,37,48–52]

and dynamic role assignment [53]. Not only could this save communication resources, it could also save energy. As the multi-agent robot is normally embedded with the digital microcontroller, which has limited resources, the research on reducing energy usage resulting from communication [54,55], actuators [56] and trajectory [57] have become recent hot topics in multi-agent research, and this area is also known as "energy aware" [58] or "energy efficient" [12,59] research.

Motivated by the aforementioned works, this study was carried out to solve the homogenous agent communication issues when finding a consensus for rendezvous applications in the broadcast and communication environment. Although a few stochastic controllers have been applied to finding a consensus for this application, the issue of communication has not been taken into consideration. Therefore, in this study, we propose an integration of an event-triggered system into the simultaneous perturbation stochastic algorithm (SPSA) and a distributed controller to obtain the minimum utilization of the channel, as well as preserving agent performances. This study can be considered an advancement in the existing method, aiming to highlight the importance of reducing energy resources from communication, which will guarantee the practicality of controllers. The effectiveness of the proposed controller was evaluated and observed in terms of trajectory, time, iteration, and number of channels taken to reach consensus for rendezvous purposes. In addition, a conventional sampling system, known as the time-triggered system, was applied in this case study as a benchmark for comparison to the event-triggered system. The obtained results are compared with traditional sampling systems in terms of channel utilization and agent performances to show the effectiveness and robustness of the proposed consensus controller. The rest of this article is structured as follows. In Section 2, the formulation of the problem is explained. Section 3 describes the system, including the global and local consensus controller designs and the proposed theorems. In Section 4, results from the simulation setup are presented and discussed, whereas Section 5 concludes this paper.

## 2. Problem Formulation

### 2.1. Preliminaries

In this section, a generic system description, covering multi-agent environment, agent control and agent connection, is presented with notations and assumptions as outlined below:

**Notations.** *Assume that R is a real number, $R_+$ is a positive real number set, and N is a non-negative integer set. The $n \times m$ zero matrix is denoted as $0_{n \times m}$. The Euclidean vector x is represented by $x_i(t) = [x_1, x_2, x_3 \ldots \ldots x_n]^T \in R^n$ with non-zero elements in Euclidean coordinates representing the positions of the agent. The agent is represented with i, neighbours with j, discrete time with t, local controller $L_i$, global controller G and broadcast signal $B \in R$. The function $f(e, x)$ contains the variables of e and x.*

**Assumption 1.** *The feedback control system presented in Figure 1 has been applied in a broadcast communication environment. The global controller G will keep broadcasting the scalar value $\beta(t)$ to agents until consensus is achieved when it reaches its desired target $x_d$. The local controller $L_i$ will determine the updated agent's state position based on the input received by the controller, which consists of random errors and deterministic errors with ET at every discrete time t. The camera will capture and update the state position of agent at G if there are changes to agent movements. This closed-loop process will repeat until the solution converges where the agent performance achievement of $P(x(t))$ is equal to 0.*

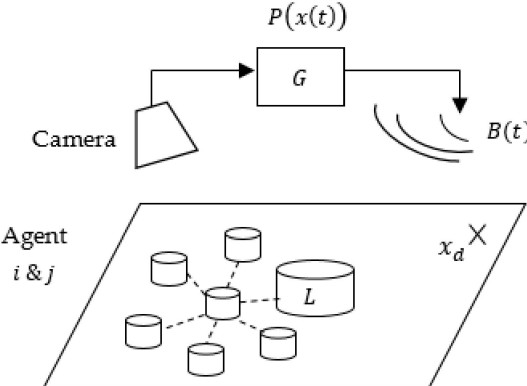

**Figure 1.** Multi-agent broadcast communication environment.

**Assumption 2.** *The physical dynamics of homogenous agent i's linear discrete time model is represented as*

$$A_i : \ x_i(t+1) = x_i(t) + u_i(t) \tag{1}$$

*where the initial state location is $x_i(t) \in R^n$ and $u_i(t)$ is the input of the controller at time t. The output of the controller, representing the next location of the state agent $x_i(t+1)$, depends on the changes in the input of the controller, also known as the local controller error.*

**Assumption 3.** *It is assumed that agent i has its own neighbor set $j \in N_i$ which is strongly linked to the radius as an undirected graph; r. r represents the radius where a relation can be formed by the agent. In undirected graph theory, $G_T = (V, E)$ where V is the vertex that represents the agent and E is the edge of each vertex representing the connection between agent i and its neighbours j or $(i, j) \in E$, as depicted in Figure 2. $N_i = \{(j|(i,j))\} \in E$ denotes a set of all the neighbors of the agent. The representation of the connection/topology of the graph and the number of edges E for each vertex V is expressed in matrix form. An adjacency matrix $A = [a_{ij}] \in R^{N \times N}$ is used to describe the graph topology, where $a_{ij} = 1$ if $(j, i) \in E$ and $a_{ij} = 0$, or otherwise. The degree matrix of the system is defined as $D = diag\{d_1, d_2 \ldots \ldots \ldots d_N\}$, where $d_i = \sum_{j \in N_i} a_{ij}$.*

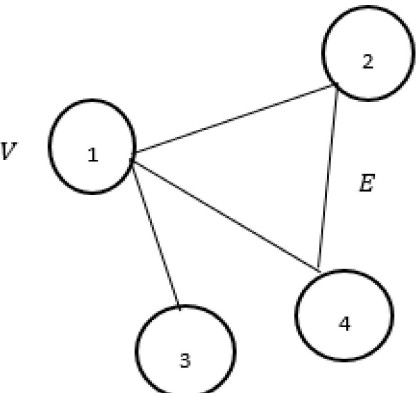

**Figure 2.** Agent *i* and neighbours *j* connected in a graph topology network.

### 2.2. Problem Formulation

Having an environment as shown in Figure 1 (Assumption 1) and a linear dynamic of agent *i* (Assumption 2) strongly connected via an undirected graph (Assumption 3), the broadcast event-triggered (BET) consensus controller is required to satisfy the motion coordination task with the rendezvous objectives function (Equation (2)) when all agents $N_i$

meet at a rendezvous point at the desired target $x_d$ (Equation (3)) with minimum utilization of resources while preserving multi-agent performances at $t$ time.

$$P(x) = \sum_{i=1}^{N} ||x_i - x_d|| \qquad (2)$$

$$lim_{t \to \infty} P(x(t)) = 0 \qquad (3)$$

Several remarks on the problem are given in the definitions below:

**Definition 1.** *The limitation of the agent to reach consensus for the rendezvous in the scope of the broadcast and communication environment. Each agent has a limited knowledge of its state position—it has the information concerning the relative position between agent i and j but does not know its position in the world's coordinates. The agent will rely on the feedback broadcasted by the G controller to determine its current position in order to the desired target point and the $L_i$ controller to determine the next updated agent's state position.*

**Definition 2.** *The communication between agent i and neighbours j in reaching an average consensus (Equation (6)) while heading to the desired target point may cause a high level of computation and utilization of communication resources. With the simultaneous perturbation stochastic algorithm (SPSA) consensus controller design, the agents' movement is determined by a local distributed controller (Equation (4)). The output of the local controller, consisting of stochastic and gradient error values, will be alternately added during even and odd times to obtain the updated agent's state position. Since communication error is taken into account during the SPSA gradient calculation, continuous data communication is thus required for the agent to meet the rendezvous point at time t.*

In order to obtain the communication error at every odd time, the agent has to exchange its state position with its connected neighbor (Equation (5)) continuously. This is to ensure that the agent moves consistently with its neighbors while avoiding the stochastic effects of SPSA during the movement. However, continuous communication by the agent and neighbors at every time-triggered interval results in a high level of computation and utilization of communication channels, especially when the target point is far from the agent and when it involves multiple agents within the environment. Without proper communication control, communication resources, i.e., the number of channels (NOC) and bandwidth of the agent will increase, which will simultaneously affect the resources used by the agent. Thus, the time, iteration and trajectory will also be affected, since there is a correlation between communication, local controller output, global controller output and the next agent's state position, as represented in Equations (1), (2), (4) and (5). Therefore, the communication factor must be taken into consideration as it will guarantee the feasibility and practicality of the controller, especially when the agent has limited power from the digital embedded microcontroller.

$$L_i: \begin{cases} \delta_i(t+1) = \alpha(\delta_i(t), B(t)) \\ u_i(t) = \beta(\delta_i(t), B(t)) + \gamma(\delta_i(t), B(t)) \end{cases} \qquad (4)$$

$$\gamma(\delta_i(t), B(t)) = k \sum_{j \in N_i} a_{ij} |x_i \delta_{i4}(t) - x_j \, \delta_{i4}(t)| \; when \; \delta_{i4}(t) \in \{1, 3, 5\} \qquad (5)$$

$$\alpha_c = \frac{1}{n} \sum_i x_i(0) \qquad (6)$$

## 3. Proposed Broadcast Event-Triggered Consensus Control Scheme

The feedback system shown in Figure 3 has been designed using an optimization algorithm known as the simultaneous perturbation stochastic algorithm (SPSA). The local controller was designed by adopting a stochastic and deterministic element of SPSA,

which will determine the linear dynamic agent's next state position $x_i(t+1)$ at time $t$. At every movement of the agent, a global controller will update the information of its state position to determine the scalar value based on the objective function calculation, which represents agent achievement. As long as the local controller receives the scalar value, it means that the agent has yet to reach the desired target. Therefore, the process will be continuous until consensus is achieved, when the termination criteria is satisfied, that is, when the agent reaches the rendezvous at the desired target point. The uniqueness of this hybrid consensus controller is that an event-triggered function (ETF) is embedded in the distributed-agent consensus controller. The ETF is specifically designed with a certain threshold value, which will limit the number of data transmissions and control updates. With event-triggered control (ETC), the value of gradient approximation of the SPSA is affected by the communication error, which has an effect on the agent's state position in the next iteration. Details of the design are explained in Section 3.1 for the global controller, in Section 3.2 for the local controller, in Section 3.3 for the event-triggered controller and in Section 3.4 for related theorems.

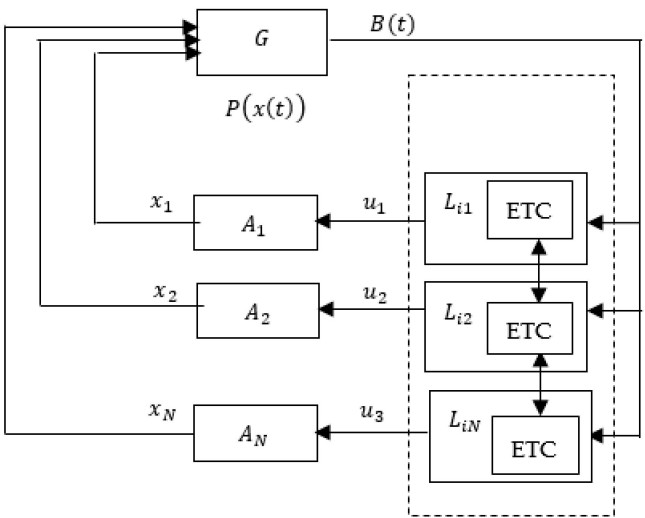

**Figure 3.** Event-triggered broadcast controller design.

*3.1. Global Controller, G*

A global controller is used to represent the agent performances in reaching consensus as to the rendezvous target given in Equation (7), where $B(t) \in R$ is the output of $G$, $x(t) \in R^{nN}$ is the state position of all agents and $P(x)$ is an objective function of rendezvous, $P(x) \in R$, where

$$G : B(t) = P(x(t)) \tag{7}$$

$$P(x) = \sum_{i=1}^{N} ||x_i - x_d|| \tag{8}$$

*3.2. Local Controller, L*

Local controller $L_i$, which is the distributed control integrated with the event-triggered process, will determine the agents' movement $A_i$ at every even or odd time $t$ is based on input from the broadcast signal, $B(t) \in R$. The local controller is denoted by Equation (11), where $\delta_i(t) \in R^v$ is the state, $B(t) \in R$ is the broadcast signal, $u_i(t) \in R^n$ is the output and $\alpha : R^v X R \to R^v$, $\beta : R^v X R \to R^v$ and $\gamma : R^v X R \to R^v$ are functions. $\alpha$ is the state function, $\beta$ is the random and deterministic function of the SPSA and $\gamma$ is the standard consensus protocol with ET. The column of vector of Equation (9) represents the variables

in the control system, where $\delta_{i1}(t)$ is the state position, $\delta_{i2}(t)$ is the broadcast signal, $\delta_{i3}(t)$ the even time and $\delta_{i4}(t)$ is the odd time.

$$\delta_i(t) = \begin{bmatrix} \delta_{i1}(t) \\ \delta_{i2}(t) \\ \delta_{i3}(t) \\ \delta_{i4}(t) \end{bmatrix} \in R^n \times R \times R \times R \tag{9}$$

$$\alpha(\delta_i(t), B(t)) = \begin{bmatrix} \Delta_i(t) \\ B(t) \\ \delta_i(t+1) \\ x_j(t) \end{bmatrix} \tag{10}$$

$$L_i : \begin{cases} \delta_i(t+1) = \alpha(\delta_i(t), B(t)) \\ u_i(t) = \beta(\delta_i(t), B(t)) + \gamma(\delta_i(t), B(t)) \end{cases} \tag{11}$$

$$\beta(\delta_i(t), B(t)) = \begin{cases} c(\delta_{i3}(t))\Delta_i & if\ \delta_{i3}(t) \in \{0,2,4,\} \\ -c(\delta_{i4}(t))\delta_{i1}(t) - a(\delta_{i4}(t))\frac{B(t)-\delta_{i2}(t)}{c(\delta_{i4}(t))} * \delta_{i1}^{-1}(t) & if\ \delta_{i4}(t) \in \{1,3,5,\} \end{cases} \tag{12}$$

$$\gamma(\delta_i(t), B(t)) = k \sum_{j \in N_i} a_{ij} \left| x_i\delta_{i4}(t) - x_j\ \delta_{i4}(t) \right| \qquad when\ \delta_{i4}(t) \in \{1,3,5\} \tag{13}$$

Based on Figure 4, the scalar value obtained from the global controller will be fed into the local controller until consensus as to the rendezvous point is achieved. The output from the local controller consists of $u_R$, determined by $\beta$ of the SPSA Bernoulli distribution error during even times and the sum of $u_{D1}$ and $u_{D2}$, obtained from $\beta$ of the SPSA gradient approximation error and $\gamma$ of the communication error, during odd times. The communication between agent $i$ and neighbor $j$ occurs only when ETF is violated. Details of the ET design are discussed in the next section. The output from $u_i(t)$ will exert an effect on the next state's position, as expressed in Equation (1).

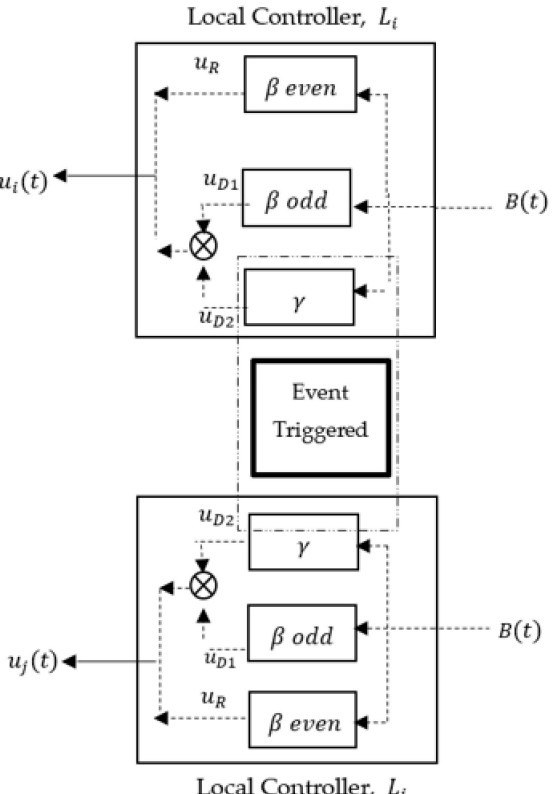

**Figure 4.** Agent local controller with event-triggered system between agent $i$ and neighbor $j$.

### 3.3. Event-Triggered Controller, $\gamma$

The event-triggered controller was designed as part of the agent distributed consensus protocol (Equation (14)) to reduce the number of communication channels while reaching an average consensus (Equation (6)) among connected agents. Since agent $i$ is connected to its neighbour $j \in N_i$ via a strongly connected undirected graph topology (Figure 2), the agent will then exchange its current state value $x_i$ with its neighbour's state value $x_j$ if it violates the ETF, which leads to the consensus of the average point (Equation (6)) and the desired target point (Equation (8)). The ETF of this system is represented by Equation (14), where $f(.)$ represents the error function, whereas $\sigma(.)$ represents the threshold function. The state measurement error (Equation (15)) depends on the difference between the agent state's position, where $t \in t_k^i, t_{k+1}^i$ denotes the event instant of agent $t_k^i (k = 1, 2, \ldots\ldots\ldots)$ and $t$ refers to the previous event. The threshold function (Equation (16)) is known as state-dependent since the value depends on the agent's state position.

The sampling process, transmission of information and updating of the controller of agents $i$ and $j$ with ET is shown in Figure 5. The sampler will first sample $n$ the state of $x_i(t)$, which becomes $x_i(nt)$ at every periodic odd sampling time $t$. Based on the given samples, the event detector will monitor the condition of the event, where the agent state of $x_i(t)$ need to be transmitted to its neighbour or not, based on Equation (14). If the event detector $i$ detects an event, $k$, in sample $n_{kj}$, the new updated sample state $x_i(n_{k+1}t)$ of agent $i$ is sent to neighbours $j$ giving sample state $x_j(n_k t)$. When neighbours $j$ receive the state sent by agent $i$, the neighbours $j$ update the agent $i$ state information and store the newly received state values of agent $i$, $x_i(nt)$. This state will then be used by the controller and event detector of agent $j$ until the next event is triggered from agent $i$. When no event occurs, this means that it fulfils the condition of Equation (14), then $x_i(n_k t)$ is directly fed into the controller, which means there is no transmission and controller update (Equation (1)) that occurs at this time. The zero-order hold will ensure that the control signal is kept constant until the next event occurs.

$$f(e_i(t), x_i(t)) \geq \sigma_i(z_i(t)) \tag{14}$$

$$e_i(t) = x_i\left(t_k^i\right) - x_i(t) \tag{15}$$

$$z_i(t) = \sum_{j \in N_i} a_{ij} \left(x_i(t) - x_j(t)\right) \tag{16}$$

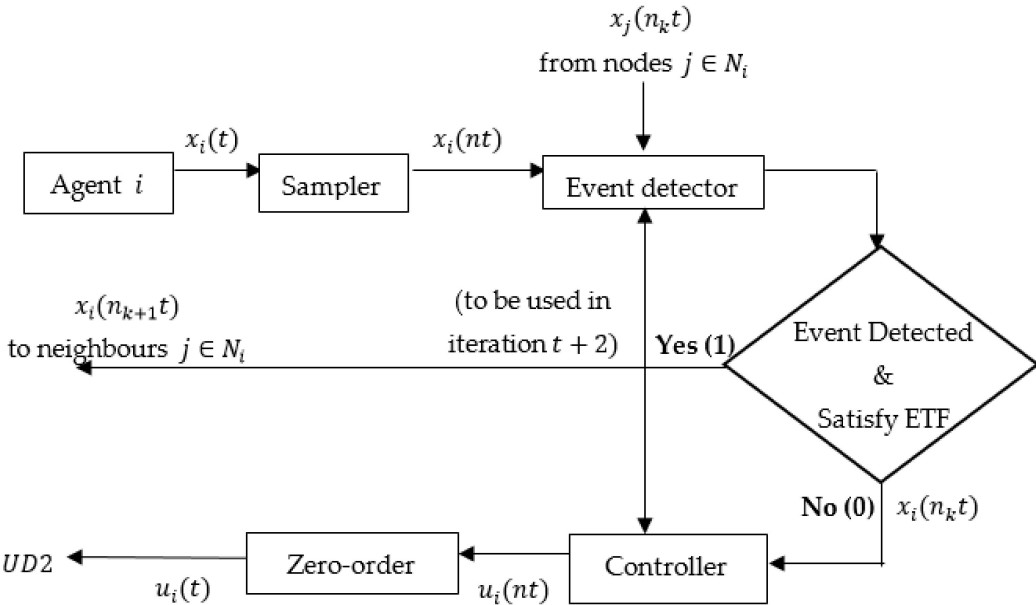

**Figure 5.** Agent *i* event-triggered controller, $\gamma$.

*3.4. Proposed Theorems*

**Theorem 1.** *Given the linear discrete system dynamics (Equation (1)) with the control input of communication as represented in Equation (16), if the system satisfies ETF (Equation (14)), where the measurement error is represented by Equation (15), the information will not be sent to the agents unless it violates the function. The information will be sent to the neighbors to update the control input. The average consensus (Equation (6)) can asymptotically marginally stabilize, which is achieved at $t \rightarrow \infty$, and reach a steady state error when the eigenvalue of Perron matrix is equal to $\lambda$, which is shown in the Gershgorin circle (Proof 1) and Lyapunoz stability, indicating $V < 0$, which is a definite negative (Proof 2).*

**Remark 1.** *The uniqueness of this ET is that the event is triggered at the periodic sampling time, which is during odd time intervals. This is known as the sampled-data ET system, where the sampled data of the agent's state position will determine the condition of the event (Equation (14)). The difference between this study and [51,52] is the integration of ET into the broadcast system, whereas in comparison with [60], the communication issue is taken into consideration for multi-agent consensus in a broadcast communication environment.*

**Remark 2.** *The threshold value $\sigma_i$ of the triggering function (Equation (14)) is state-dependent, where the agent's state position depends on its neighbor's state position.*

**Assumption 4.** *The network of topology is assumed to be strongly connected among inter-agents under the undirected graph to achieve a marginally stable consensus.*

**Proof of Theorem 1.** Refer to Appendix A. □

**Theorem 2.** *For BET communication, suppose that the objective function P is given and assumed that P is changing satisfying $\nabla P(x_d) = 0$ with ETF, $f(e, x)$ (Equation (14)) applied in the agent and neighbors' communication system using a standard distributed protocol (Equation (16)). Let $L_i$ and G be given by Equations (7), (8) and (11). If the BET satisfies the condition of Theorem 1 and Lemma 1, then $\lim_{t \rightarrow \infty} x(t) = x_d$ w.p.1 is achieved.*

**Proof of Theorem 2.** Refer to Appendix B. □

## 4. Results and Discussion

In this section, a series of simulations that include ten connected homogeneous agents (Assumption 3) were assumed to be initially located at the planar coordinates (refer to Table 1). With optimal SPSA parameter settings and ET settings, as presented in Table 2, in a feedback control system in a broadcast and communication environment (Assumption 1), the agent was expected to reach consensus and meet at a rendezvous point at the desired target point located at (80, 80), as shown in Figure 6. The SPSA gain of $a_k$ and $c_k$ depended on Equations (17) and (18). The agent worked based on the "objectives" function determined in Equation (2) and the termination criteria, based on the measurement state error value, which should reach zero. The results obtained specifically in relation to the efficiencies of the time and iterations of this research were based on results obtained using MATLAB 2015B simulation software with the Intel® Core™ i5-2410M CPU @ 2.3GHz processor, running on a 64-bit operating system.

$$a_k = a/(t+1+A)^{\alpha_1} \qquad (17)$$

$$c_k = c/(t+1)^{\gamma_1} \qquad (18)$$

**Table 1.** Agent state position.

| Agent | 1 | 2 | 3 | 4 | 5 | 6 | 7 | 8 | 9 | 10 |
|-------|-----|------|------|-------|------|-------|-------|------|-------|-------|
| Agent position | 5,5 | 10,5 | 5,10 | 10,10 | 15,5 | 15,10 | 15,15 | 20,5 | 20,10 | 20,15 |

**Table 2.** Optimal simultaneous perturbation stochastic algorithm (SPSA) and event-triggered (ET) parameter settings for the broadcast event-triggered (BET) controller.

| SPSA Parameters | Values |
|:---:|:---:|
| $a$ | 0.2 |
| $A_1$ | 30 |
| $\alpha_1$ | 0.6 |
| $c$ | 1 |
| $\gamma_1$ | 0.06 |
| $\sigma_i$ | 30 |

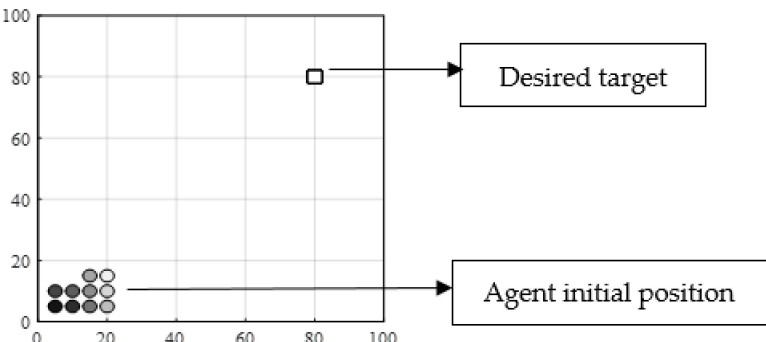

**Figure 6.** Agent 2D workspace (initial state position and desired target).

The evaluation of agent performances in terms of time, iteration, trajectory and number of channels taken by the multi-agent to reach the rendezvous point in a few time runs were then recorded. This started with evaluation of the BET consensus controller, and the BTT consensus controller was then observed in order to compare the effectiveness of the

proposed controller with the conventional sampling system. The results are divided into two sections, starting with the BET and followed by the performance comparison of the BET and BTT in finding the consensus in relation to the desired rendezvous target.

### 4.1. Performances of the BET Consensus Controller

#### 4.1.1. Time and Iteration

The average time the agent took to reach the rendezvous point for the desired target point in 10 time runs was an average of 84.3413 s and 676.6 iterations as shown in Table 3 (Proof of Theorem 2 and Proof 4). The shortest time and iterations taken by an agent to converge was at least 35.996 s with 312 iterations when the state measurement error, which indicates the performance index, showed a reading of 0 after a certain number of iterations as shown in Figure 7.

**Table 3.** Average time and iteration.

| No. | Time (s) | Iteration |
|-----|----------|-----------|
| 1 | 87.831 | 724 |
| 2 | 106.991 | 708 |
| 3 | 35.996 | 312 |
| 4 | 104.732 | 854 |
| 5 | 74.954 | 636 |
| 6 | 46.307 | 408 |
| 7 | 78.563 | 669 |
| 8 | 104.44 | 850 |
| 9 | 109.76 | 892 |
| 10 | 92.839 | 713 |
| Average | 84.2413 | 676.6 |

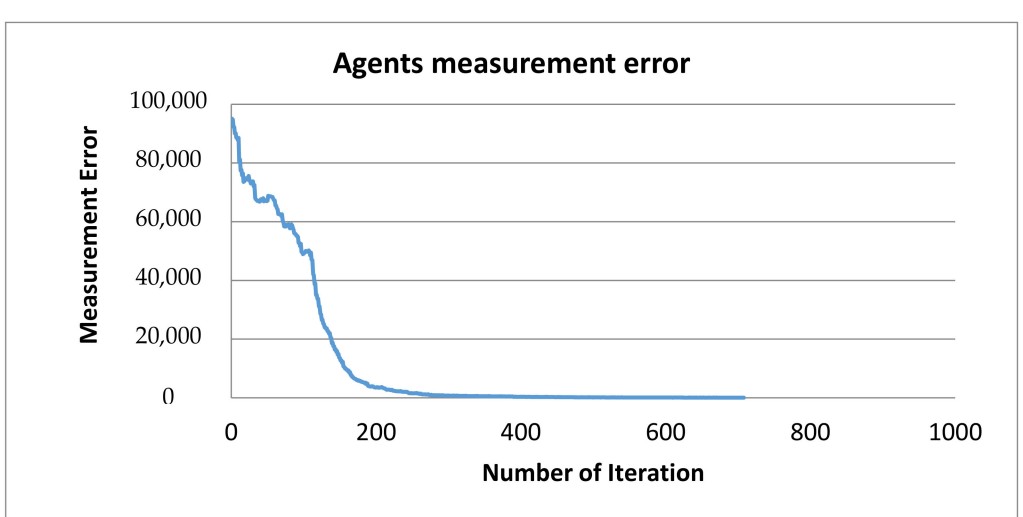

**Figure 7.** Agent measurement error.

#### 4.1.2. Trajectory of Each Agent

Figure 8 shows the agent's movement obtained from BET, demonstrating that the rendezvous was reached at 225.424 s with 1104 iterations. The first 300 iterations showed that the agents moved in a scattered manner and were not too close with one another. However, when it reached more than 500 iterations, the communication error caused the

agents to meet at the average consensus point (proving Theorem 1) while heading to the goal point. Figure 8 shows that the agent's movement was synchronized, with a reduction of error.

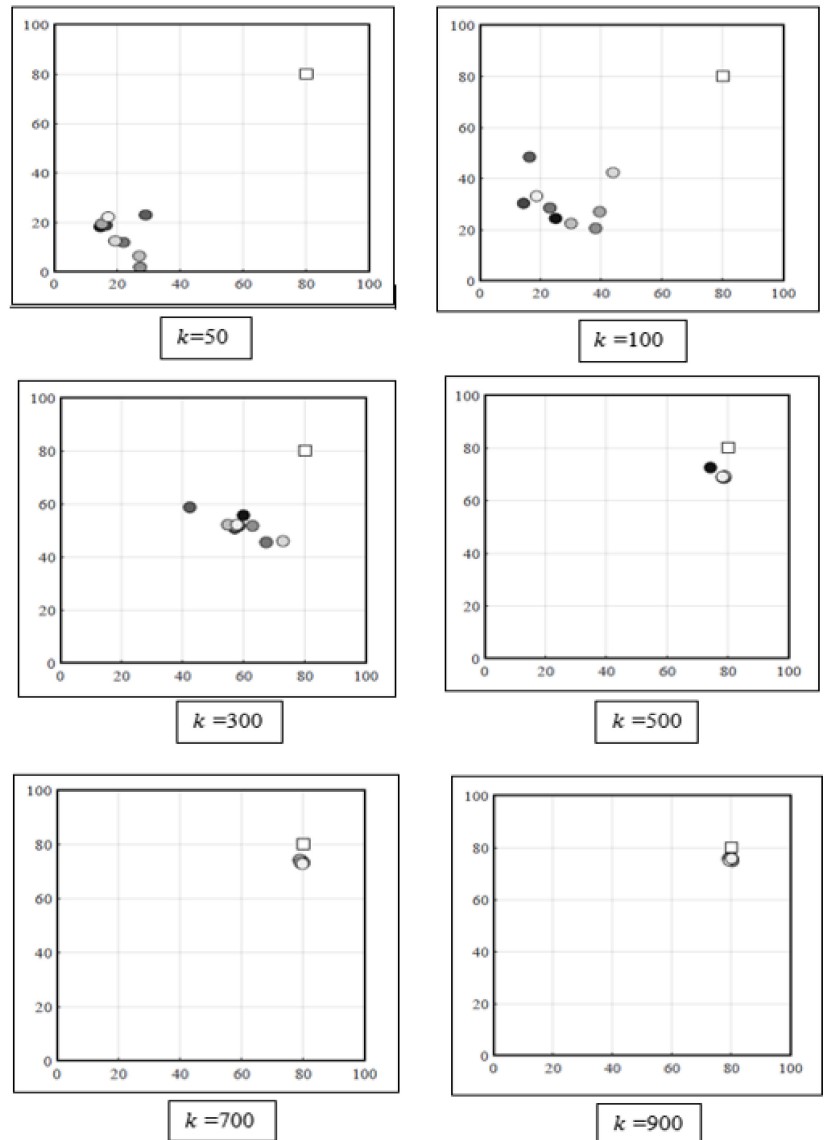

**Figure 8.** Agent movement to meet at rendezvous point.

### 4.1.3. Utilization of Channels

The channel usage was recorded as shown in Table 4 for the BET controller. The number of channels (NOC) was not constant during ET, and it depended on the value of ETF whether to stop or to allow agent communication among agents. Based on Table 4, the agent successfully reduced its channel utilization by 21.79%. Details of the calculations of the agent are presented below.

**Table 4.** Number of channels per 100 iterations. NOC, number of channels.

| Time (s) | 20.4 | 40.8 | 61.2 | 81.6 | 102 | 122.4 | 142.8 | 163.2 | 183.6 | 204 | 225.2 |
|---|---|---|---|---|---|---|---|---|---|---|---|
| Iteration | 100 | 200 | 300 | 400 | 500 | 600 | 700 | 800 | 900 | 1000 | 1104 |
| NOC | 248 | 181 | 177 | 255 | 436 | 500 | 500 | 500 | 500 | 500 | 520 |

Total of iteration = 1104/2 = 552 iterations.
Time per iteration = 225.424/1104 = 0.204 s.
Each iteration is equivalent to 10 channels, which carried a total of 552 × 10 = 5520 channels.

### 4.2. Comparison of BET and BTT Agent Performances

#### 4.2.1. Time and Iteration Efficiencies

The average readings of time and iteration taken by the multi-agent system to reach consensus in relation to the rendezvous in ten time runs are shown in Table 5. BET was proven to lead BTT by 57.153 s, with 177 iterations. The efficiency of BET and BTT in reaching consensus depended on the effectiveness of the broadcast consensus controller in working with the distributed controller with either event sampling or conventional time sampling.

**Table 5.** BTT and BET time and iteration values.

| Environment | BTT | | BET | |
|---|---|---|---|---|
| Variables | Time | Iteration | Time | Iteration |
| Average | 141.394 | 853.9 | 84.241 | 676.6 |

#### 4.2.2. Agent Trajectory

The trajectory or agent movement with BET and BTT were observed at every 50 iterations to estimate the trajectory patterns obtained by both controllers. With BET, the 10 agents' movement was scattered during the first 200 iterations, before the agents showed movement heading to the rendezvous point when they reached 200 to 900 iterations. The movement can be seen clearly in Figure 9, in which the agents' state position accumulates within the range of $x$–$y$ coordinates when it reaches $80 < x < 100$ and $70 < y < 90$. This was unlike the BTT system, in which the agents' movement looked consistent, as illustrated in Figure 10. The agents were pulled among one another (Assumption 3) to reach an average consensus while heading to the rendezvous point.

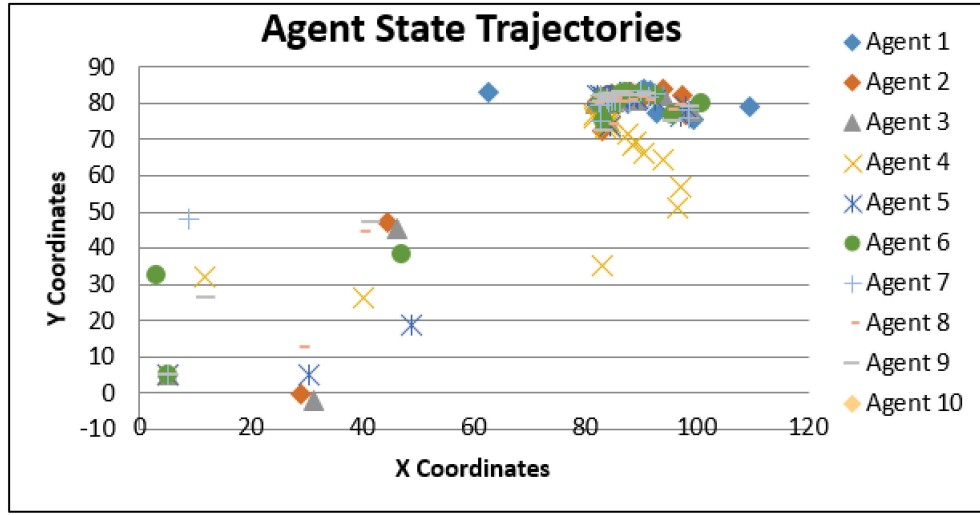

**Figure 9.** Agent state trajectories (BET).

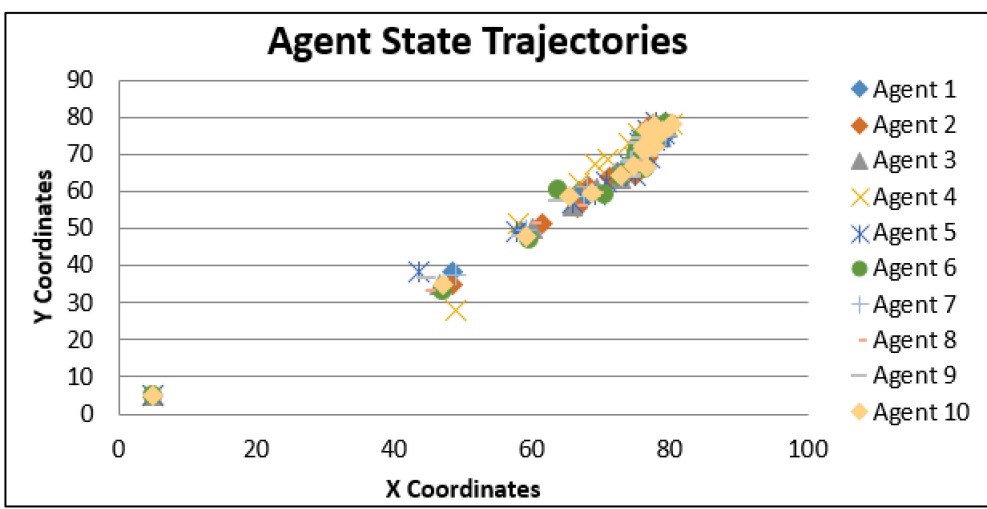

**Figure 10.** Agent state trajectories (BTT).

### 4.2.3. Utilization of Channels

Tables 6 and 7 show the NOC used by BET and BTT in a selected time run. The BET met the rendezvous target at 225.2 s with 1104 iterations, whereas BTT converged at 100.8 s with 840 iterations. The NOC was recorded per 100 iterations to observe how the NOC was involved in communication between the agent and neighbors along the way until the agent reached the rendezvous point. BET showed a total of 71.09% NOC usage as compared to BTT, which utilized 100% of the channels. This was due to the effect of implementing event-triggering in the BET system, which resulted in the usage of channels which were not full for the first 500 iterations, as shown in Figure 11. As a result, there were at least 28.9% usage of channels reserved for the agent with the use of the BET controller as compared to none of the channels being reserved in the BTT system.

**Table 6.** Number of channels per 100 iterations (BET).

| Time (s) | 20.4 | 40.8 | 61.2 | 81.6 | 102 | 122.4 | 142.8 | 163.2 | 183.6 | 204 | 225.2 |
|---|---|---|---|---|---|---|---|---|---|---|---|
| Iterations | 100 | 200 | 300 | 400 | 500 | 600 | 700 | 800 | 900 | 1000 | 1104 |
| NOC | 248 | 181 | 177 | 255 | 436 | 500 | 500 | 500 | 500 | 500 | 520 |

**Table 7.** Number of channels per 100 iterations (BTT).

| Time (s) | 12 | 24 | 36 | 48 | 60 | 72 | 84 | 96 | 100.8 |
|---|---|---|---|---|---|---|---|---|---|
| Iterations | 100 | 200 | 300 | 400 | 500 | 600 | 700 | 800 | 840 |
| NOC | 500 | 500 | 500 | 500 | 500 | 500 | 500 | 500 | 200 |

The NOC of the first 10 iterations were recorded to observe the patterns of agent communication at each iteration. Referring to Table 8 and Figure 11, there were only a few agents that would send the information of the agent's state position at each iteration, which depended on the ETF condition concerning whether there were any violations. As a result of this, the channel utilization can be reduced in the BET system as compared to the BTT system, which fully utilized the channel, as shown in Table 9 and Figure 12.

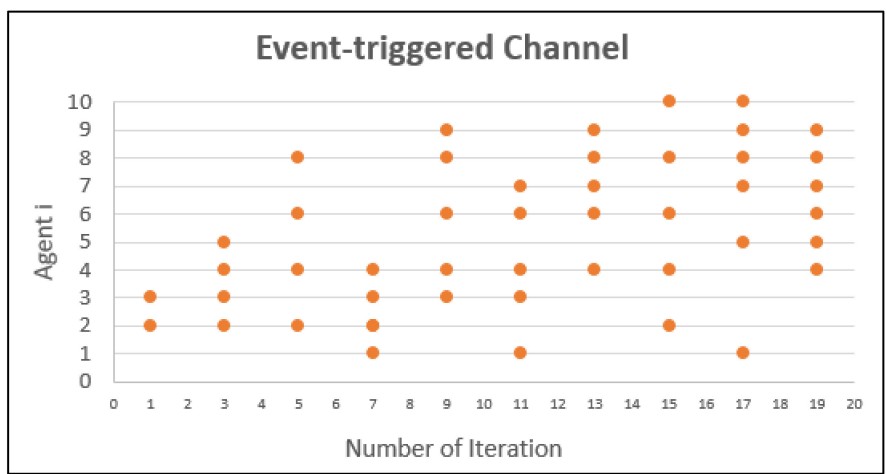

**Figure 11.** NOC per iteration (BET).

**Table 8.** Number of channels per 10 iterations (BET).

| Time (s) | 0.2 | 0.61 | 1.02 | 1.43 | 1.84 | 2.24 | 2.65 | 3.06 | 3.47 | 3.88 |
|---|---|---|---|---|---|---|---|---|---|---|
| Iterations | 1 | 3 | 5 | 7 | 9 | 11 | 13 | 15 | 17 | 19 |
| NOC | 2 | 4 | 4 | 5 | 5 | 5 | 5 | 5 | 6 | 6 |

**Table 9.** Number of channels per 10 iterations (BTT).

| Time (s) | 0.12 | 0.36 | 0.6 | 0.84 | 1.08 | 1.32 | 1.56 | 1.8 | 2.04 | 2.28 |
|---|---|---|---|---|---|---|---|---|---|---|
| Iterations | 1 | 3 | 5 | 7 | 9 | 11 | 13 | 15 | 17 | 19 |
| NOC | 10 | 10 | 10 | 10 | 10 | 10 | 10 | 10 | 10 | 10 |

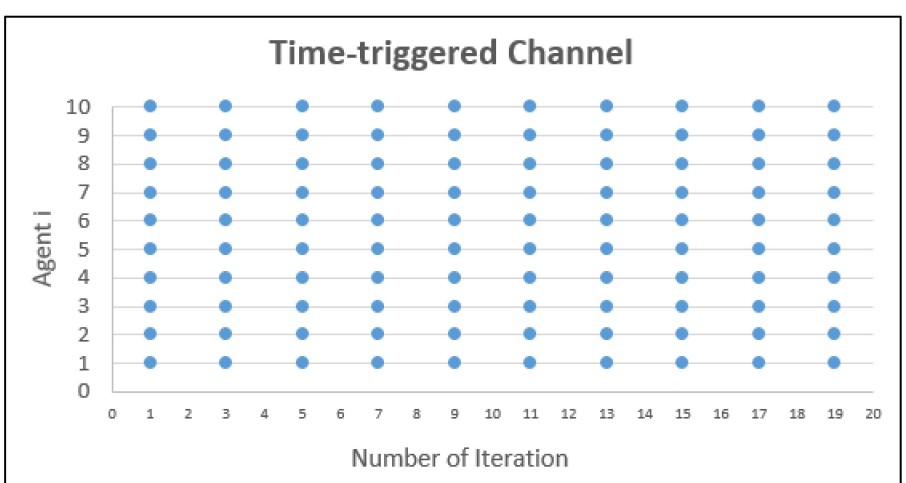

**Figure 12.** NOC per iteration (BTT).

## 5. Conclusions

A hybrid controller in a broadcast and communication environment with the BET system produced very promising results in terms of finding a consensus as to the rendezvous point. The agent was able to reach consensus regarding the rendezvous for the desired target in a minimum time and number of iterations, while reducing the utilization of channels. A lower number of channels will reduce communication resources, which will simultaneously reduce the number of control updates and save energy. In fact, BET

was much better than BTT in terms of time, iterations, number of channels and trajectory movement. Furthermore, the trajectory of agents showed less communication effects towards the gradient value of SPSA, which would have apparent effects on the agent's next state position. Thus, BET has been proven to be efficient and practical in order to find a consensus as to a rendezvous point in a broadcast and communication environment.

In future, this research can be validated using a real multi-agent robot which uses broadcast and distributed control supported by a network wireless system. The effectiveness and robustness of the proposed controller to achieve convergence with probability one, w.p.1 is expected from the results obtained in our modeling. Additional physical tests can be conducted to measure agent efficiencies by recording the time, channel utilization and energy usage. In addition, instead of reducing the communication channels to limit the usage of communication resources, an extension of this research towards energy-aware and energy efficient systems can be proposed. To this end, the total energy obtained from communication and motion can be calculated, which will solve the issue of limitations in agent resources in agent microcontrollers. Finally, the idea of embedding event-triggered systems in other complex systems and in relation to various dynamics of agents is recommended in order to guarantee the practicality and viability of multi-agent controllers.

**Author Contributions:** Conceptualization, N.S. and Z.H.I.; methodology, N.S. and Z.H.I.; software, N.S; validation, N.S.; formal analysis, N.S. and Z.H.I.; investigation, N.S. and Z.H.I.; resources, N.S. and Z.H.I.; data curation, N.S.; writing—original draft preparation, N.S.; writing—review and editing, N.S. and Z.H.I.; visualization, N.S. and Z.H.I.; supervision, Z.H.I.; project administration, N.S. and Z.H.I.; funding acquisition, Z.H.I. All authors have read and agreed to the published version of the manuscript.

**Funding:** This work was supported in part by the Ministry of Higher Education Malaysia and Universiti Teknologi Malaysia under Grant no. R.K130000.7843.5F348 and R.K130000.7643.4C355.

**Institutional Review Board Statement:** Not applicable.

**Informed Consent Statement:** Not applicable.

**Data Availability Statement:** No new data were created or analyzed in this study. Data sharing is not applicable to this article.

**Acknowledgments:** Thanks to UTM for providing their available software and robotic platform.

**Conflicts of Interest:** The authors declare no competing interests.

## Appendix A

**Proof 1 of Theorem 1.** With the ET model in the system, in which the state measurement error is expressed as Equation (15), the system dynamic (Equation (1)) is represented as Equation (A1),

$$
\begin{aligned}
x_i(t+1) &= x_i(t) + u_i(t) \\
&= (\hat{x}_i(t) - e_i(t)) + u_i(t) \\
&= (\hat{x}_i(t) - e_i(t)) + (\sigma\, a_{ij}(\hat{x}_j(t) - \hat{x}_i(t))) \\
&= -e_i(t) + \sigma\, a_{ij}\hat{x}_j(t) + (1 - \sigma\, a_{ij})\hat{x}_i(t)
\end{aligned}
\tag{A1}
$$

Based on the normalized Laplacian matrix, Equation (1) is represented by a Perron matrix in discrete time as Equation (A2), where

$$
\begin{aligned}
x_i(t+1) &= x_i(t) + u_i(t) \\
&= x_i(t) + \frac{1}{1+d_i} a_{ij} \sum_{j \in N_i} (x_j(t) - x_i(t))
\end{aligned}
$$

since $a_{ij} \sum_{j \in N_i} (x_j(t) - x_i(t))$ is equivalent to $-Lx(t)$

$$
\begin{aligned}
&= x_i(t) + \tfrac{1}{1+d_i}(-L\, x(t)) \\
&= x_i(t) + (-(I+D)^{-1})L\, x(t)) \\
&= x_i(t) - (I+D)^{-1})L\, x(t)) \\
&= I - (I+D)^{-1}(I+A)x(t) \\
&= (I+D)^{-1}(I+A)x(t) \\
&= P_e x(t)
\end{aligned}
\tag{A2}
$$

Substituting Equation (15) into Equation (A2), where $x_i(t_k^i) = \hat{x}_i(t)$ is the latest broadcast state

$$
x(t+1) = P_e(e_i(t) + x_i(t))
\tag{A3}
$$

If the conditions of Equation (14) are satisfied, there is no event occurring at this time with the error of Equation (15), and $e_i(t)$ is equal to 0, which means that there is no transmission and control update happening with the agent and its neighbours. This means that the sampled data are not updated, and the agent's state value will remain the same. The opposite occurs for the situation where the conditions of Equation (14) are violated. In this scenario, an event occurs, in which the error is equal to Equation (15) and the new state position of Equation (A3) is sent to the connected neighbors for a control update. Even though the state measurement error gives a significant effect to the next agent's state, as stated in Equation (A2), the results of the closed loop system are guaranteed asymptotically to reach average consensus, Equation (6), among agents in the group when the eigenvalues of square matrix $P$ are strictly contained in the Gershgorin circle criterion. The eigenvalues of the Perron matrix for a topology of 10 agents are strongly connected with an undirected graph—these are $\lambda_1 = 0$, $\lambda_2 = 0.5 \pm 0.5i$ and $\lambda_{10} = 1$. The eigenvalues inside the Gershgorin circle unit are expected to be $\lambda_{10} = 1$. $\square$

**Proof 2 of Theorem 1.** Consider a Lyapunov function (Equation (A4)), which is represented by Equation (A5) when $Lx \triangleq y = [y_1, y_2, \ldots . y_N]^T$.

$$
Vx(t) = \frac{1}{2}x(t)^T Lx(t)
\tag{A4}
$$

$$
\begin{aligned}
\dot{V} &= \frac{\partial V}{\partial x} \cdot \frac{dx}{dt} = x^T L\dot{x} \\
&= -x^T L(Lx + Le) \\
&= -LLxx^T - LLex^T \\
&= -y^T y - y^T Le
\end{aligned}
\tag{A5}
$$

With the Laplacian matrix, Equation (A5) will be

$$
\begin{aligned}
\dot{V} &= \sum_i y_i^2 - \sum_i \sum_{j \in N_i} y_i(e_i - e_j) \\
&= \sum_i y_i^2 - \sum_i |N_i| y_i e_i + \sum_i \sum_{j \in N_i} y_i e_j
\end{aligned}
\tag{A6}
$$

For $a > 0$, $\dot{V}$ can bound using inequality $|xy| \leq \frac{a}{2}x^2 + \frac{1}{2a}y^2$, thus

$$
\dot{V} \leq -\sum_i y_i^2 + \sum_i a|N_i| y_i^2 + \sum_i \frac{1}{2a}|N_i| y_i^2 + \sum_i \sum_{j \in N_i} \frac{1}{2a} y_j^2
\tag{A7}
$$

Since the communication graph is undirected,

$$
\sum_i \sum_{j \in N_i} \frac{1}{2a}e_j^2 = \sum_i \sum_{j \in N_i} \frac{1}{2a}e_i^2 = \sum_i \frac{1}{2a}|N_i|\, e_i^2
$$

Therefore,

$$\dot{V} \leq -\sum_i (1 - a|N_i|) y_i^2 + \sum_i \frac{1}{a} |N_i| e_i^2$$
$$\dot{V} \leq \sum_i (\sigma_i - 1)(1 - a|N_i|) y_i^2 \tag{A8}$$

which shows that the system is stable when the derivative of the Lypunov function is negative definite for $0 < \sigma_i < 1$. □

**Lemma 1.** *Each state position of the agent, $x_i$ is sampled at every specific odd time, t bounded by $t + 1$. The channel utilization in each sampling time is limited to a maximum number of agents $N_i$ per sampling, which shows that Zeno behavior does not exist and can be excluded in this case, as proven in Proof 3.*

**Proof 3 of Theorem 1.** The agent deterministic movement (Equation (11)) is determined by the sum of the ET consensus control input (Equation (13)) with the SPSA deterministic input of the local distributed agent controller, which occurs during odd time intervals (Equation (12)). The agent movement is alternated with the stochastic movement at even times. □

**Appendix B**

**Proof 1 of Theorem 2.** The following facts prove the theorem:

(1)   The dynamics of the agent according to Equations (A9)–(A11) are equivalent to the simultaneous perturbation stochastic algorithm and the sequences of agent movements during $(x(0), x(1), x(2) \ldots \ldots (t \to \infty)$ and converge to w.p.1, which is under the conditions of (A1) to (A7).

(2)   The relation holds for even time $t \in \{0, 2, 4, \ldots \ldots\}$ and odd time $t \in \{1, 3, 5, \ldots \ldots\}$ as per Equations (A9) and (A10), which will also affect the next agent state as stated in Equation (A9).

$$B(t) = P(x(t)) \tag{A9}$$

$$B(t + 1) = P(x(t)) + (c(t) + \Delta\_i(t)) \tag{A10}$$

(3)   For $t \in \{0, 2, 4, \ldots \ldots\}$, the relation between BET represented by Equation (A11) is held.

$$x(t + 2) = x(t) - \left[ a(t) \frac{B(t + 1) - B(t)}{c(t)} * \Delta_i^{-1}(t) \right] - k \sum_{j \in N_i} a_{ij} |x_i - x_j| \tag{A11}$$

Substitution of Equations (A9) and (A10) into Equation (A11) will obtain Equation (A12), which can be simplified to Equation (A13).

$$x(t + 2) = x(t) - a(t) \frac{(P(x(t) + c(t)\Delta_i(t)) - P(x(t)))}{c(t)} \Delta_i^{-1}(t)$$
$$- k a_{ij} \sum_{j \in N_i} (x_i - x_j) \tag{A12}$$

$$x(t + 2) = x(t) - a(t) d(x(t), \Delta_i(t), c(t)) - k a_{ij} \sum_{j \in N_i} (x_i - x_j) \tag{A13}$$

By applying Taylor's theorem to Equation (A10), the expected value of $d(x(t), \Delta_i(t), c(t))$ is near to the gradient of $P(x(t))$, as shown in Equation (A14).

$$E(d(x(t), \Delta_i(t), c(t)|x(t)) = \nabla P(x(t) + O(c(t))(c(t) \to 0) \tag{A14}$$

Thus, the agent state of $x(t + 2)$ is represented by Equation (A15), in which the gradient descent method has been integrated with the sum of the agent communi-

cation error between the connected agents. With these, $x(t)$ will converge to a local minimum point of $P$.

$$E((x(t+2)|x(t)) \cong x(t) - a(t) P(x(t)) - k \, a_{ij} \sum_{j \in N_i} (x_i - x_j) \tag{A15}$$

(4) The collective dynamics movements of agent BET at $|x(1) - x(0)|$, $|x(3) - x(2)|$, $|x(5) - x(4)|, \ldots\ldots$ and $|x(n+1) - x(n)|$ show that the agents will converge to $x_d$ or 0 w.p.1. The dynamics are determined with the aim of asymptotically achieving consensus locally and globally, as highlighted in Theorems 1 and 2.

(5) If (ET1), (ET2), (A3) and (A4) hold, the dynamics in Equation (1) will converge to a minimum point of function $P$, represented by the quadratic function (Equation (8)) that measures the degree of achievement when the agents move from the initial state $x_0$ to the target point $x_d$ w.p.1. The settings of broadcast (A3) and (A4) and event-triggered (ET1) and (ET2) are imposed for tuning $L_i$ and $G$ as defined in Equations (7) and (11).

- (ET1) $0 < k < 1/N$, then $\lim_{t \to \infty} x_i(t) - x_j(t) = 0$
- (ET2) $f(e_i(t), x_i(t)) \leq \sigma_i(z_i(t))$, then $e_i(t) = 0$, else $e_i(t) = x_{\hat{i}(t)} - x_{i(t)}$

(6) If the conditions of SPSA convergence satisfy (A1)–(A7) and the conditions of ET satisfy (ET1) and (ET2), the system will converge when all agents reach the desired target $x_d$.

- (A1) $F$ is twice differentiable.
- (A2) $x^*$ is an asymptotically stable equilibrium of the gradient system $\dot{z}(\tau) = -\nabla P(z(\tau))$, where $\tau \, \epsilon R_{0+}, z(\tau) \in R^{nN}$, and the stability is in the Lyapunov sense.
- (A3) $a(t)$ and $c(t)$ for every $t \in \{0, 2, 4, \ldots\}$, $\lim_{t \to \infty} a(t) = 0, \sum_{t=0}^{\infty} a(t) = \infty, \lim_{t \to \infty} c(t) = 0$, and $\sum_{t=0}^{\infty} (a(t)/c(t))^2 < \infty$.
- (A4) $\Delta_{i1}(t)$, $\Delta_{i2}(t)$, $\ldots\ldots\ldots\ldots\Delta_{iN}(t)$ are random variables from a probability distribution which is continuous and symmetric about zero.
- (A5) $E[P(x(t) + c(t)\Delta(t)^2]$ is bounded for all $t \in N$,
- (A6) For a compact set $S \subseteq R^{nN}$ such that $\dot{z}(\tau) = -\nabla f(z(\tau))$ with $x(0) \in S$ results in $x(\infty) = x^*$, $x(t) \in S$ occurring infinitely often for almost all the sample points of $\Delta_i(t)(i = 1, 2, \ldots N)$ and $(t = 0, 1, 2 \ldots \ldots .)$
- (A7) $sup_{t \in N}||x(t)|| < \infty$ w.p.1, then $\lim_{t \to \infty} x(t) = x^*$. □

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
