# Peer review of "Broadcast Event-Triggered Control Scheme for Multi-Agent Rendezvous Problem in a Mixed Communication Environment"

_applsci, doi:10.3390/app11093785_

Round 1

Reviewer 1 Report

General comments:

Paper needs to be properly formatted to ensure that all diagrams, text and equations are within the boundaries of the document.

Avoid the over use of brackets i.e (Equation (6)) > (Equation. 6) 

The language and grammar is rather difficult to understand in certain places. I recommend that the paper is proof read throughout.

A number of tables i.e (table 5,7,8,9) have overlap which makes it impossible to read.

Questions.

1.

You did these tests in MATLAB, did you consider applying your model in a system like ROS and evaluating variable conditions that may be more representative of real world conditions? 

If you assume perfect input/output, then your model may not be able to function with the same level of confidence as in a virtual test environment.

This is particularly import when designing a multi-agent system to traverse pathways to a rendezvous point.

Section 3.

You have a lot of equations in section 3, although I appreciate that you are trying to show proof of theorems, it may be work doing most of the working in an appendix and just have the main equations in the body of the text. This is an optional recommendation to improve readability. 

5. Conclusion. 

Your conclusion is very short and acts more of a summary to the points that are covered in the main body and abstract. It would be beneficial if you could include potential real-world applications for your model to improve the current state-of-the-art. 

Further to this I would recommend a future work section that would include the physical tests of the system to measure how the robots function in the real-world. I understand that in the current climate this may be difficult but at the moment there are lots of studies been conducted in virtual environments that do not really convey to real-world challenges in robotics/ai.

You have gone down the mathematical route to justify your approaches (and correctly so) but the practical application of the system beyond that. It would be nice to see some consideration of this factor in the paper. 

Otherwise this is a solid piece of research. 

Author Response

We thank the Reviewer #1 for the constructive comments.

Please see the attachment for our response.

Reviewer 2 Report

The paper addresses an important topic in multiagent coordination, especially in relation to kinetic agents, and potentially this would be very interesting to readers. However it needs major rewriting in order to reach its full potential. 

Items 63 and 64 don't seem to be referenced at all in the text, yet they seem to be amongst the most interesting relevant works. The introduction mentions other approaches to solving the consensus rendezvous problem but it is unclear in what respect this study is a complement/advancement/improvement /alternative to such approaches, hence the originality and contribution of this work is not fully explicit. As there are other event triggered  control strategies in the literature (for example Dong 2020, 16th International Conference on Control and Automation, IEEE) I do not fully understand in what respect this work is different/better/complementary etc.

I would also recommend a few remarks in the concluding section about the potential of the proposed approach for practical applications, which might help the reader to appreciate the significance of the broader idea and not get lost in the details of the mathematical formulations and proofs (which are very nice, but may restrict the appeal to a broader multiagent systems audience).

Author Response

We thank the second reviewer for constructive comments.

Please see the attachment for our response.
